# Intermediate-Level Diet Quality of Brazilian Paralympic Athletes Based on National and International Indexes

**DOI:** 10.3390/nu15143163

**Published:** 2023-07-17

**Authors:** Willian V. D. Schneider, Carolina A. L. Sasaki, Teresa H. M. da Costa

**Affiliations:** Graduate Program in Human Nutrition, Department of Nutrition, University of Brasilia, Brasília 70910-900, DF, Brazil; wvagners@hotmail.com (W.V.D.S.); carolinanutricionista09@gmail.com (C.A.L.S.)

**Keywords:** parathletes, nutrition assessment, diet quality

## Abstract

Diet quality indexes are used to characterize the dietary patterns of individuals and populations. The objective of this study was to compare two specific diet quality indexes, namely the Brazilian Healthy Eating Index Revised (BHEI-R) and the Global Diet Quality Score (GDQS), among Brazilian parathletes. This comparison was performed using either the initial 24 h recall (Rec1) or an assessment of usual dietary intake. Additionally, our study aimed to explore the association of these indexes with sociodemographic and behavioral sport variables. This cross-sectional, observational study evaluated 101 disabled athletes, including 23 international-level and 78 regional-/national-level participants, with a distribution of 82 males and 19 females across 13 Paralympic modalities. The Multiple Source Method (MSM) was employed, utilizing data from two or four non-consecutive 24 h food recalls. The comparison between the Rec1 and the assessment of usual dietary intake revealed the following median (IQR) values: for the BHEI-R, they were 60.3 ± 11.1 and 80.7 ± 6.2, respectively; for the GDQS, they were 19.5 ± 6.5 and 18.3 ± 2.6, respectively. Most athletes had diets classified as either “in need of modification” (according to BHEI-R) or of “moderate risk” (according to GDQS). The comparison between type of sport (team/individual), age, sex, income, education, sport scholarship, and nutritional support between the diet quality indexes is presented. Athletes involved in individual sports exhibited higher scores than team sports for BHEI-R (*p* < 0.02), and athletes receiving nutritional support achieved higher scores on both indexes (*p* < 0.03). The analysis of diet quality using the initial Rec1 with the BHEI-R was deemed sufficient to evaluate the diet quality of these athletes. However, when evaluating sporadically consumed food groups, the adoption of GDQS is necessary to assess usual dietary intake. We found that both BHEI-R and GDQS can be utilized to evaluate the diet quality of athletes with disabilities, and the diet quality of parathletes reached an intermediate score level.

## 1. Introduction

Diet quality indexes incorporate nutritional criteria and recommendations based on food intake, eating habits, and cultural environment in order to classify the diet of individuals and populations [1,2]. The information the indexes provide allows for the simultaneous analysis of the following parameters: adequate intake level of nutrients, amounts of portions and grams consumed, and number of different types of food consumed [3,4].

The Index Nutrients (IN) was the first index, proposed by Jenkins and Guthrie in 1984 [5]. Over time, different indexes have been proposed. Kennedy and collaborators created the Healthy Eating Index (HEI) to better understand the patterns in the food consumed by the population in the United States of America (USA) [6].

The Brazilian Healthy Eating Index-Revised (BHEI-R) comprises 12 dietary items and is used to assess the diet quality of Brazilians [3], whereas the Global Diet Quality Score (GDQS), a research initiative by the Intake Institute (https://www.intake.org), addresses 25 different aspects of the diet, which are assigned positive, negative, or total scores. The GDQS can be used to compare different countries and cultures regarding the risk of noncommunicable diseases [4].

Studies of diet quality indexes in athletes with disabilities are scarce and data are needed for comparing and evaluating the nutritional behavior of these athletes’ diets. Only one study on the diet quality in Brazilian athletes with disabilities was found after a careful database search. The study evaluated 20 athletes during a training camp using the BHEI-R [7]. 

Food consumption studies often have limitations. One common limitation is that some studies focus on food consumption during a specific period, such as a 7-day training camp [7,8,9]. However, this type of consumption does not accurately reflect an individual’s intake when they are at home. At home, meals are typically prepared in larger quantities, resulting in leftovers that may be consumed during subsequent meals or the following day. As a result, consecutive days of intake tend to have a high correlation, which can negatively impact the accuracy of dietary assessment [10].

The present study aimed to determine the BHEI-R, and GDQS diet quality indexes in Brazilian parathletes, adopting the first 24 h recall or the usual intake with 2 or 4 nonconsecutive days of intake, and associate the findings with sociodemographic and behavioral variables pertaining to sport. One previous study has evaluated BHEI_R in parathletes during a training camp and observed a moderate-quality diet [7]. Additionally, our previous results have shown some micronutrient inadequacies in the parathletes’ diets [11]. Therefore, we tested the hypothesis that the diet quality of the parathletes in their home environment might be low for both indexes, reflecting a poor and/or unhealthy diet.

## 2. Materials and Methods

This cross-sectional, observational study occurred between September 2018 and August 2019. All adult (>18 y) athletes with disabilities living in the Federal District (FD), which includes Brazil’s capital Brasília, were invited to participate in the study. Eligible participants were male and female athletes from all Paralympic sports who participated in regional, national, or international events; all agreed to participate. Athletes with intellectual and severe visual disabilities, for whom there is no standardized 24 h recall available, as well as novice athletes with disabilities, were excluded from the study.

We evaluated 101 disabled athletes from 13 Paralympic sports. The athletes provided demographic, sport category, and socioeconomic data. The data was collected over a 12-month period in which athletes and sports were split into four groups; interviews with each group were done sequentially over a 3-month period. The collection scheme secured that athletes of all types of sports would have a dedicated training period, competition calendar, and seasonality represented during the collection year. 

The ethics committee of the University of Brasília School of Health Sciences approved this study (Protocol n.2.502.000 CAAE 79851917.1.0000.0030) and all athletes signed a written informed consent declaration.

### 2.1. Dietary Intake

The food consumption was obtained through the 24 h recall, which requested information on the food, drinks, and supplements consumed on the previous day [10]. All athletes answered two nonconsecutive 24 h food recalls and, a third and fourth 24 h recall were randomized and applied to 50% of the athletes in each group. The first 24 h recall (Rec1) was at the training site, home, or in the office and, the other recalls were done via phone call. To collect the 24 h recall data, the five-step multiple pass method (MPM) was employed [12]. Additionally, a support kit consisting of utensils (cups, glasses, and cutlery) and a food portion photography guide [13] were utilized. The MPM consisted of five stages. The first stage involved the uninterrupted listing of the foods and beverages consumed in the past 24 h. The second stage was based on a list of commonly forgotten foods for each meal. In the third stage, the interviewee provided the consumption time for each food/supplement or beverage, detailing the location and occasion of consumption. The fourth stage, known as the elaboration stage, required the interviewee to provide additional details about the meals, including quantities of the consumed foods, brands, and preparation methods. At this point, household measures such as teacups, cups, tablespoons, teaspoons, etc., were recorded. In the fourth stage, information regarding the meals time and occasion were reviewed. The fifth and final stage was the final review, where a thorough examination of the already reported foods was conducted, and any possible forgotten or unreported foods were included [12]. 

The information from the 24 h recall was registered in food portion size and units and converted into grams of consumed foods using the standardized quantities from the photography guide [13] and a Brazilian household measurement table [14]. To quantify the content of food groups and dietary items from the food recall data, the Nutrition Data System for Research software (NDSR, version 2018) developed by the Nutrition Coordinating Center at the University of Minnesota, USA, was utilized [15]. Brazilian food items and recipes were either directly entered in the NDSR or adapted from foods with similar nutritional profiles [16].

### 2.2. Usual Intake

To minimize random error from 24 h recall, we used methodologies to account for day-to-day dietary variability. The usual intake was performed for both methods (GDQS and BHEI-R) by collecting the four 24 h recalls. To calculate the usual intake, we adopted the Multiple Source Method (MSM) (Department of Epidemiology of the German Institute of Human Nutrition Potsdam-Rehbrücke, 2008–2011) [17]. In short, the MSM applies a three-step methodology to correct intra-individual intake errors. The first step considers the intake probability using a logistic regression calculation. The second step takes the values of the observed intake group data from each 24 h recall. A linear regression model was applied, and the values were modeled, and the corresponding residuals of the linear regression transformed to normality/symmetry by a two-parameter Box–Cox transformation. Then, the inter- and intra-individual variance were determined, and the intra-individual variance removed, which resulted in a shrinkage of the residuals distribution. The quantity calculated in this shrinkage process for each person was then transformed back to the original scale. The third step, which was the distribution of usual intake value for the study population, was estimated by multiplication of the results of steps 1 and 2 [17,18].

### 2.3. Diet Quality Assessment

#### 2.3.1. Brazilian Healthy Eating Index Revised (BHEI-R)

We used BHEI-R that was adapted from HEI-2005 for the Brazilian population (Appendix A). The BHEI-R ranges from 0 to 100, and is composed of 12 components, which include nine food groups from the Brazilian Food Guide 2006, two nutrients (sodium and saturated fat), and SoFAAS (energy from solid fat, alcohol, and added sugar). The portions of the nine food groups and sodium are presented as energy density (g/1000 total energy intake), whereas saturated fat, and SoFAAS are presented as a percentage of total energy intake (TEI). The scoring values for the nine food groups from the Brazilian Food Guide have a minimum zero score for no consumption. However, a minimum score is given for intake above the established threshold for saturated fat, sodium, and SoFAAs. The calculation of intermediate values was proportional to the amount consumed [3].

The BHEI-R classification is set in three levels, in which less than 51 points are classed as a poor diet, between 51 to 80 points as needing modification, and over than 80 points is classed as a healthy diet [19].

#### 2.3.2. Global Diet Quality Score (GDQS)

The GDQS is a metric for scoring diet quality (Appendix A) related to the risk of noncommunicable diseases. The metric was composed of 25 food groups (range between 0 to 49), and a point value was assigned based on the observed range of consumption in grams per day. Each food group is scored using three scoring ranges, except for high-fat dairy, which uses four ranges. The red meat food group was scored only with an intermediate value of 9–46 g/day. The GDQS was obtained by summing all food group point values; the classification has three levels: below 15; 15 to 23; and >23, meaning high, moderate, and low risk of noncommunicable diseases, respectively [20]. 

### 2.4. Statistical Analysis

Data organization was performed in Microsoft 365’ Excel. The descriptive variables were sex, education level, age, socioeconomic status, nutritional support, sport ranking level, and scholarship program, stratified by the BHEI-R and the GDQS. The BHEI-R calculation was done using a template developed by Previdelli et al. [3] in the Stata software (version 10.0). The GDQS calculation was done in Microsoft 365’ Excel. The estimated usual intake of the BHEI-R and the GDQS was done with MSM to reflect the long-term average intake of the nutrient or food group [17]. The calculations of the diet quality indexes were performed separated by Rec1 and usual intake. The normality test was done with the Shapiro–Wilk test. Comparison between the Rec1 and the usual intake was performed with the Wilcoxon signed-rank test. Comparison between groups for each descriptive variable was performed with the Mann–Whitney test and the three levels of the athlete’s education level were compared with the Kruskal–Wallis test followed by the Bonferroni–Dunn post hoc test for nonparametric independent variables. The correlations were calculated between the diet quality indexes, GDQS and BHEI-R, with the Spearman’s rho test for the nonparametric variables. The statistical software package, SPSS, for Windows version 20 was used to perform the analysis. We considered the statistical significance at *p* < 0.05 for all analyses.

## 3. Results

The protocol for the present study with Brazilian parathletes used four repeated recalls from nonconsecutive days over one year, which allowed us to compare the diet quality indexes BHEI-R and GDQS between Rec1, the 24 h recalls’ mean value (original), and the usual intake values. The usual intake distribution presents a lower or similar kurtosis and a lower variation when compared to the original uncorrected distribution. During the shrinkage process, extreme values in the score/index distribution are adjusted. Specifically, the lower tail (5th percentile) is increased, while the higher tail (95th percentile) is decreased [21,22]. This process is further supported by the reduction in standard deviation (SD) observed in the usual values compared to the original and Rec1 values. To account for measurement errors, we applied corrections to each food group or dietary item in both analyzed indexes. Subsequently, the usual values were used to calculate the final usual score for BHEI-R and GDQS (Figure 1).

Figure 1 shows the distribution of the BHEI-R and GDQS with Rec1, original, and usual intake. The construction of the GDQS presents a longer list of food groups and metrics for low and high consumption, in order to identify potential inadequacies of the extremes of the intake distribution, which is not possible to do with the BHEI-R.

The scoring format of GDQS assigns points ranging from 0 to 2 or 4 for each food group, resulting in a broader range of scores. In contrast, BHEI-R utilizes a scoring format with fewer food groups, assigning points ranging from 0 to 5, 10, or 20 (Appendix A). This variation in scoring methods could potentially explain the greater dispersion of points observed for BHEI-R compared to GDQS (Figure 1).

We performed the Spearman correlation between GDQS and BHEI-R for usual intake and Rec1, and both generated significant positive correlations. The correlation between the BHEI-R usual and GDQS usual was ρ = 0.579; *p* < 0.0001, between the “BHEI-R and Rec1” and “GDQS and Rec1” was ρ = 0.484; *p* < 0.0001. The plots shown in Figure 2 demonstrate the shrinkage of the distribution of usual intake, resulting in a narrower spread of the usual scores compared to the Rec1 scores.

The correlation results prompted us to compare the outcomes between both scores using either the usual score or the Rec1 score. 

Table 1 presents the BHEI-R and GDQS classification with usual intake or Rec1. None of the athletes reached a value of a healthy diet with the BHEI-R, with their usual intake. The median usual intake total score for BHEI-R was significantly higher than the Rec1 and the median usual intake total score for GDQS was significantly lower than the Rec1. The diets of most athletes were classified as needing modification and moderate risk, with BHEI-R and GDQS, respectively. The GDQS positive value with usual intake was higher than that with Rec1 (Appendix A).

The parathlete population of the FD were mostly male (82%) and team sports athletes (56%). They were mostly 30 years old or older (64%), had a low income (67%), and had secondary education (high school) (43%). Most parathletes participated at regional and national levels (78%), and the majority received no sports scholarship (55%), and no nutritional support (70%). 

Table 2 shows the sum of ranks for the score considering usual intake and the Rec1 for the sociodemographic and behavioral sport variables according to GDQS and BHEI-R. The GDQS scores were significantly lower for usual intake than for the Rec1 for male, individual sports, for 30 years or older athletes, with low and high income, having or not sports scholarship, receiving nutritional support, with tertiary education or equivalent, and with international, and at a regional/national sport ranking level.

When comparing the sum of ranks for the scores of BHEI-R between usual intake and Rec1 values, we found that the usual intake had significantly higher values than the Rec1, in specific subgroups. These subgroups included male, team sports participants, athletes aged 30 years or older, and athletes who did not receive nutritional support (refer to Table 2).

We performed the Kruskal–Wallis Test between primary, secondary, and tertiary education levels. The result obtained with the GDQS for the analysis of Rec1 show that secondary education had a lower score than tertiary education or equivalent education (*p* = 0.02) (Appendix A).

We performed the Mann–Whitney test comparing sport types (team or individual sports) and nutritional support (yes or no). We found that with BHEI-R for Rec1 the individual sports scored higher than the team sports. Additionally, for both GDQS for Rec1 and BHEI-R for Rec1 and usual intake, the group that received nutritional support scored higher than the group that did not receive nutritional support. The other comparisons between groups were not significant (Appendix A).

## 4. Discussion

This study included all 101 adult parathletes from 13 Paralympic sports who fulfilled the inclusion criteria and live in the Federal District of Brazil. To the best of our knowledge this is the first study to compare different methods for evaluating food intake and diet quality indexes in this population group. 

We found that depending on the study aim, considering only one Rec and using the BHEI-R diet quality methodology is sufficient to analyze the total diet quality of Brazilian athletes with disabilities. However, if the interest is in evaluating sporadically consumed food groups such as fish or the overconsumption of food groups, the usual intake with the GDQS is a better methodology to be applied. 

The mean of one 24 h recall from a group of individuals captures frequently consumed foods with good precision, but only the usual intake can capture the long-term intake [21,22] The methodology of assessing usual intake, which involves obtaining at least two non-consecutive 24 h recalls from a subset of individuals, increases the likelihood of capturing sporadically consumed food items. Additionally, this approach allows for the correction of intraindividual variations in dietary intake.

The shrinkage of the intake distribution (Figure 1) allows the correct analysis of the distribution of extreme values [21]. Our method, which involved collecting dietary intake data from parathletes over a one-year season using two or four days of intake, provides valuable insights that contribute to the existing knowledge and offer guidance for selecting appropriate methodologies for evaluating dietary quality.

The BHEI-R and GDQS differs in their scoring indexes. The BHEI-R accounts for the energy density of food groups whereas the GDQS accounts for food intake in grams. This difference is crucial in athletes with disabilities because of a high variation in the distribution of total energy intake (TEI). Athletes who have suffered a spinal cord lesion, for example, have lower energy expenditure of around 14 to 27% [11,23]. Consequently, the amount of food consumed should be lower to prevent unwanted weight gain that may directly impact the quality score of the diet. Therefore, the GDQS is less sensitive to detect this variation and fluctuation of results. This argument is corroborated when we contrasted the highest total score observed among the parathletes for the GDQS, which was 37.2 out of 49 (76%), while the BHEI-R was 86.7 out of 100 (87%) of the maximum score.

In our previous study of micronutrient assessment in the parathletes of the Federal District we found a high prevalence of inadequate intake of vitamin D, calcium, magnesium, vitamin A, vitamin C, and other micronutrients [11] This led us to raise the hypothesis tested in this study that the parathletes of the Federal District of Brazil would have a low-quality diet. However, we found that they had an intermediate-level diet quality score, for both the GDQS and the BHEI-R indexes. The rejection of our initial hypothesis may be due to the fact that diet quality involves different aspects that may better relate to non-communicable disease outcome than to low micronutrient intake.

The Athlete Diet Index (ADI) was published and validated concomitantly during the period of our field work [24]. The ADI has 68 closed questions split in three sections that assess usual intake, training load, and demographic details. The scoring is of 125 points divided over 90 for high score, between 66 to 89 for a medium score, and under 65 for a low score [24]. Unfortunately, the protocol we developed could not be adapted to apply the ADI to our data because the ADI requires information on training load, which was not collected in this study. 

The BHEI-R has been used with the Brazilian Paralympic track and field athletes [7] Dietary data was recorded by photography on four consecutive days of food intake during a week-long evaluation at a sport center. The results of this study show that female athletes were not statistically different (n = 7; 63.7 ± 5.9) than males (n = 13; 61.3 ± 5.3) for the BHEI-R score and that both sexes need to improve their diets. Data collection was done away from the athletes’ homes, which limits any inference on habitual intake [7]. 

We found that athletes with disabilities with higher education levels also had the highest BHEI-R [25]. A qualitative study with the general population observed that individuals with higher education consume a higher variety of food types [26]. When we compared the diet quality score between individual and team sports, we found that individual sports had the BHEI-R score higher than team sports. One explanation for this finding is that athletes who compete in individual sports are mainly and solely responsible for the result outcome, whereas in team sports the outcome responsibility is split among all, and dietary choices may also play a role on the outcome. It is important to point out that eating habits of team sports athletes is less investigated and fewer studies are available on the subject than those on individual sports athletes [27]. However, some evidence suggests that the diet quality of team sports athletes is lower than that reported for athletes involved in individual sports [28]. This topic is still unresolved because in another study using the ADI, team sports athletes scored higher than individual sports athletes [24].

Athletes who had nutritional support had a higher sum of rank for diet quality score than athletes who did not report such support, but the same was not observed between international sport level athletes when compared to national/regional level. This result reinforces the need for and importance of athlete’s nutritional support to achieve a better diet quality. In an evaluation of diet quality in non-athlete Brazilians, the same result was observed where individuals who received nutritional support obtained a higher score for diet quality compared with individuals without nutritional support [29]. Evaluation of ADI for athletes training fewer hours (0–11 h/week) and longer hours (12 h/week), showed that athletes who train longer may be at risk of a poor dietary pattern, but we did not obtain such difference when considering that international level athletes might train longer hours, and be more focused on and committed to the sport [24].

The limitations of the study were that we could not cover all modalities of Paralympic sports and our protocol did not allow the use of the ADI to assess athletes’ diet quality using this instrument. The strengths were the sample size, the analysis of the usual intake of Paralympic athletes, and the evaluation of the diet quality using two different methods. 

Altogether, our results expand our knowledge on the subject and give a broader view of diet quality for parathletes from different modalities.

## 5. Conclusions

We conclude that both BHEI-R and GDQS can be used to evaluate the diet quality of athletes with disabilities following the perspectives defined in the study. The use of one recall is preferable when the group of athletes is large and the objective is to assess the quality of the whole diet, because of the cost and applicability. The adoption of usual intake is indicated to collect data on food consumption on at least two non-consecutive days, with any sample size of athletes if the objective is to investigate sporadically consumed food groups, such as fish, and whole grains. Finally, each index and analysis has both advantages and disadvantages. The final decision is defined by the investigation aims. We have concluded that nutritional support favors a better diet quality score for athletes with a disability in the Federal District, Brazil.

## Figures and Tables

**Figure 1 nutrients-15-03163-f001:**
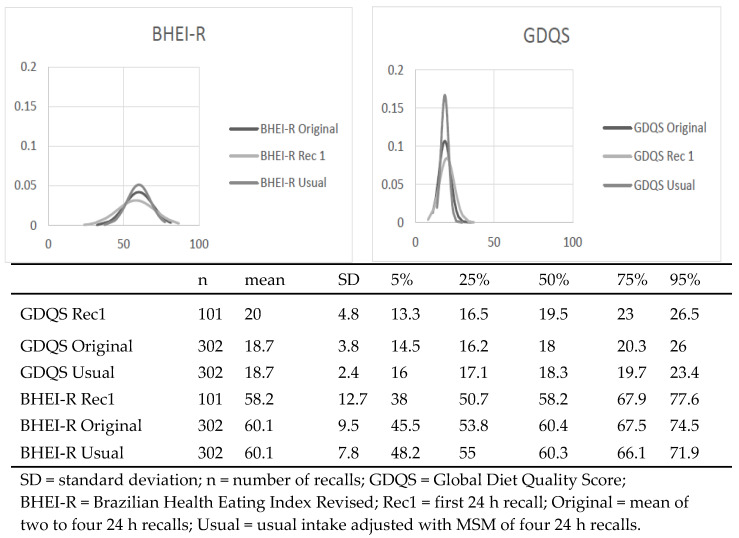
Descriptive analysis and density plots for the Global Diet Quality Score (GDQS) and the Brazilian Healthy Eating Index Revised (BHEI-R) total score values of the first 24 h recall (Rec1), mean of four day of intake (original) and corrected (usual intake) data for measurement error due to intra-person variation with Multiple Source Method (MSM) from 101 athletes with disabilities Brasília/Brazil, 2018–2019.

**Figure 2 nutrients-15-03163-f002:**
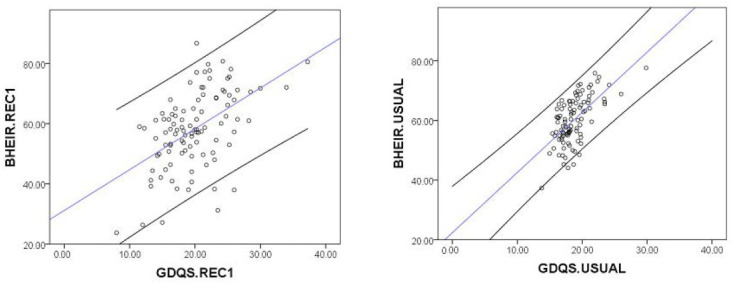
Dispersion plot of Brazilian Healthy Eating Index Revised (BHEI-R) versus Global Diet Quality Score (GDQS) for usual intake (usual) or first 24 h recall (Rec1). Blue line = trend line and black lines = 95% confidence interval.

**Table 1 nutrients-15-03163-t001:** Descriptive statistics according to the score of the Brazilian Healthy Eating Index Revised and Global Diet Quality Score of the usual intake and the first 24 h recall from 101 athletes with disabilities Brasília/Brazil, 2018–2019.

Indexes	Classification	n	Usual/Rec1	Median	IQR	Minimum	Maximum	*p*
BHEI-R Classification	Inadequacy	11	Usual	48.29	4.36	37.30	59.55	
		25	Rec1	44.01	10.90	23.71	60.98	
	Need Modification	90	Usual	61.31	10.39	45.24	77.56	
		73	Rec1	61.40	12.39	41.16	79.75	
	Healthy Diet	-	Usual	-	-	-	-	
		3	Rec1	80.72	6.19	80.51	86.70	
	Total	101	Usual	60.33	11.15	37.30	77.56	0.04
		101	Rec1	58.18	17.16	23.71	86.70	
GDQS Classification	<15	2	Usual	14.37	1.22	13.76	14.98	
		12	Rec1	13.38	2.00	8.00	14.75	
	15–23	93	Usual	18.18	2.56	15.42	22.60	
		64	Rec1	19.25	3.63	15.00	23.00	
	>23	6	Usual	23.82	2.53	23.39	29.86	
		25	Rec1	25.25	2.25	23.25	37.25	
	Total	101	Usual	18.26	2.64	13.76	29.86	
		101	Rec1	19.50	6.50	8.00	37.25	0.001

n = athletes with disabilities; Rec1 = First 24 h recall; IQR = interquartile; Usual = calculation of usual intake in software MSM (2008–2011) with all 24 h recalls; Wilcoxon signed-rank test with usual intake and Rec1 in the total classification.

**Table 2 nutrients-15-03163-t002:** Socio-demographics and sport-related support presented as number and the sum of rank from 101 athletes with disabilities from 13 Paralympic sports stratified by the Global Diet Quality Score (GDQS) and the Brazilian Healthy Eating Index Revised (BHEI-R) according to the usual intake or the first 24 h recall (Rec1)—Brasília/Brazil, 2018–2019.

			GDQS			BHEI-R		
Characteristics		n	Usual	Rec1	*p*	Usual	Rec1	*p*
Sex	Male	82	1092.00	2311.00	0.000	2208.00	1195.00	0.02
	Female	19	50.00	140.00	0.07	87.00	103.00	0.75
Sport	Individual	45	230.00	805.00	0.000	502.00	533.00	0.86
	Team	56	620.00	976.00	0.15	1136.00	460.00	0.01
Age	18–30 y	37	315.00	388.00	0.58	331.00	372.00	0.76
	30 y+	64	469.00	1611.00	0.000	1480.00	600.00	0.00
Income	Low	67	728.00	1550.00	0.01	1398.00	880.00	0.11
	High	34	154.00	441.00	0.01	364.00	231.00	0.26
Sports scholarship	No	55	490.00	1050.00	0.02	934.00	606.00	0.17
	Yes	46	315.00	766.00	0.01	667.00	414.00	0.17
Nutritional support	No	70	1058.00	1427.00	0.28	1598.00	887.00	0.04
	Yes	31	52.00	444.00	0.000	272.00	224.00	0.64
Education	Primary education	24	101.00	199.00	0.16	171.00	129.00	0.55
	Secondary education	43	353.00	593.00	0.15	593.00	353.00	0.15
	Tertiary education or equivalent	34	120.00	475.00	0.002	381.00	214.00	0.15
Ranking Level	International	23	42.00	234.00	0.003	175.00	101.00	0.26
	Regional/National	78	1061.00	2020.00	0.02	1858.00	1223.00	0.11

n = athletes with disabilities; Rec1 = First 24 h recall; Usual = calculation usual intake in software MSM with four 24 h recalls; Wilcoxon signed-rank test with usual intake and Rec1.

## Data Availability

The data are not publicly available due to privacy. All data is presented clearly and honestly.

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
