# Peer review of "Intermediate-Level Diet Quality of Brazilian Paralympic Athletes Based on National and International Indexes"

_nutrients, 2023, doi:10.3390/nu15143163_

Round 1

Reviewer 1 Report

Dear Authors

As one of the reviewers, I express my personal scientific opinion on your work. I would like to reassure you that I was trying to be positive and constructive but particularly as fair and honest as possible to your work. The clear explanation provided in Method’s section is appreciated. I should also note that the originality of the study, the statistical approach used and the work done on tables and figures are all positive points. However, the lack of the calculation of the Effect Size and CI are somewhat negative points.

Please accept my judgment with a positive and constructive way.

General comments:

1.      In my point of view, the title of the study does not strongly support the whole work done; since you did not only examine the diet quality intake of the athletes but also you evaluated several other characteristics (Table 2) of the athletes. In other words, you have done much more than what your title implies.

2.      Although the article is in general well-understandable and well-presented, it is written in slightly an informal and not such an academic way. I would like to suggest you to try to revise the whole paper accordingly for improving as much as possible the written academic language.

3.      Could you please check grammar and syntax throughout the manuscript for clarity?

Abstract:

4.      Lines 17-18: “The comparison between the Rec1 and usual intake with the Wilcoxon Signed Ranks Test and the correlation test with the Spearman rho test”. The verb is missing.

Introduction:

5.      Lines 59-60: You hypothesized that the diet quality of the para-athletes may be low for both indexes reflecting poor and/or unhealthy diet. However, how did you reach this hypothesis. No strong relevant evidence is provided into your Introduction section. Consequently, based on what evidence you have reached this hypothesis?

Methods:

6.      Lines 63-64: You reported: “All adults (>18y) athletes… were invited”. The athletes invited to do what?

7.      You nicely evaluated the quality assessment of the foods consumed by the athletes. Since the foods that were consumed by the athletes were not weighted in grams, what about the quantification of the foods consumed?

Results:

8.      Figure 1: Since the data were non-parametric, in my point of view it should be better to present Median and IQR, instead of mean+/-SD. Unless the data presented in fig 1 were all parametric.

9.      Table 1: I think the abbreviation SD is not need here.

Discussion:

10.  Line 244: Please keep space between [20] and In…

11.  Lines 242-246: Please consider to elaborately moved this info to your Introduction section for better justifying your hypothesis (this is related to point 5).

12.  Lines 254-255: You reported: “Unfortunately, the protocol we developed could not be adapted to apply the ADI to our data”. What do you actually mean? Could you please explain better?

13.  Line 275 (a sample related to point 2 above): It could be better to replace the word “But” with the word “However,”.

Moderate editing of English language required.

Author Response

  1. In my point of view, the title of the study does not strongly support the whole work done; since you did not only examine the diet quality intake of the athletes but also you evaluated several other characteristics (Table 2) of the athletes. In other words, you have done much more than what your title implies.

A: Yes, we thank you very much. It was a very detailed job. But in the elaboration of the title our proposal was to be as objective as possible.

  1. Although the article is in general well-understandable and well-presented, it is written in slightly an informal and not such an academic way. I would like to suggest you to try to revise the whole paper accordingly for improving as much as possible the written academic language.

A: We have made a complete language review to make it more suitable for reading a scientific paper.

  1. Could you please check grammar and syntax throughout the manuscript for clarity?

A: Yes. We did.

Abstract

  1. Lines 17-18: “The comparison between the Rec1 and usual intake with the Wilcoxon Signed Ranks Test and the correlation test with the Spearman rho test”. The verb is missing.

A:  Lines 18-22: The comparison between the first 24-hour recall (Rec1) and the assessment of usual dietary intake revealed the following median (IQR) values: for the final Brazilian Healthy Eating Index Revised (BHEI-R), they were 60.3±11.1 and 80.7±6.2, respectively; for the final Global Diet Quality Score (GDQS), they were 19.5±6.5 and 18.3±2.6, respectively.

Introduction

  1. Lines 59-60: You hypothesized that the diet quality of the para-athletes may be low for both indexes reflecting poor and/or unhealthy diet. However, how did you reach this hypothesis. No strong relevant evidence is provided into your Introduction section. Consequently, based on what evidence you have reached this hypothesis?

A: Lines 64-67. The evidence from a previous research was included to support our hypothesis. The text now reads: One previous study evaluated BHEI_R in parathletes during a training camp and observed a moderate quality diet [7]. Also, our previous results had showed some micronutrient inadequacies in the parathletes' diets [11]. Therefore, we tested the hypothesis that the diet quality of the parathletes in their home environment might be low for both indexes, reflecting a poor and/or unhealthy diet.

Methods

  1. Lines 63-64: You reported: “All adults (>18y) athletes... were invited”. The athletes invited to do what?

A: The text was adjusted.

  1. You nicely evaluated the quality assessment of the foods consumed by the athletes. Since the foods that were consumed by the athletes were not weighted in grams, what about the quantification of the foods consumed?

A: We have included an explanation in the methods to make it clear. Lines 111-114.

The information from the 24-hour recall was registered in food portion size and units and converted into grams of consumed foods using the standardized quantities from the photography guide [13] and a Brazilian household measurement table [14].

We also include below a detail description of our methods of data collection to make it clear for the reviewer.  The process of data collection was as follows. In the R24h, the athlete reports which foods, supplements, and beverages have been consumed in the past 24 hours, along with their respective time, location, preparation method, and quantities in household units. For obtaining the quantity information, the utensil sample (cutlery and cups) and an illustrative book of food portions (Crispim et al., 2017; Zabotto et al., 1996) were utilized. The Multiple Pass Method (MPM) technique was employed for administering the R24h. This method, developed and adopted by the United States Department of Agriculture (USDA) and known as MPM, consists of five stages (Conway et al., 2003). The first stage involves the uninterrupted listing of the foods and beverages consumed in the past 24 hours. The second stage is based on a list of commonly forgotten foods for each meal. In the third stage, the interviewee provides the consumption time for each food/supplement or beverage, detailing the location and occasion of consumption. The fourth stage, known as the elaboration stage, requires the interviewee to provide additional details about the meals, including quantities of the consumed foods, brands, and preparation methods. At this point, household measures such as teacups, American cups, tablespoons, teaspoons, etc., are recorded. In the fourth stage, it is also necessary to review certain information regarding the time and occasion. The fifth and final stage is the final review, where a thorough examination of the already reported foods is conducted, and any possible forgotten or unreported foods are included. The data collection and entry stages for the R24h occurred simultaneously, optimizing the processing time for dietary data. Before entering the data into the program, the household measurement data was converted into grams of consumed foods. This conversion utilized the utensil sample, the illustrative book of food portions, and the household measurement table from the Family Budget Survey (POF) 2008-2009 (Crispim et al., 2017; IBGE. Instituto Brasileiro de Geografia e Estatística & Coordenação de Trabalho e Rendimento, 2011; Zabotto et al., 1996).

Results

  1. Figure 1: Since the data were non-parametric, in my point of view it should be better to present Median and IQR, instead of mean+/-SD. Unless the data presented in fig 1 were all parametric.

A: The data in Figure 1 presents the application of the Multiple Source Method (https://nugo.dife.de/msm). The values correspond to the first and second moment of the distribution (mean and SD). The distribution of intake values is transformed to normality/symmetry by a two parameter Box-Cox transformation. We removed the third and fours moment (kurtosis and skewness) for clarity.

  1. Table 1: I think the abbreviation SD is not need here.

A: SD was removed from the abbreviation list

Discussion

  1. Line 244: Please keep space between [20] and In…

A: It has been adjusted.

  1. Lines 242-246: Please consider to elaborately moved this info to your Introduction section for better justifying your hypothesis (this is related to point 5).

A: It was included in the introduction.

Lines 48-52. The evidence from a previous research was included to support our hypothesis. The text now reads: Studies of diet quality indexes in athletes with disabilities is scarce and data are needed for comparing and evaluating the nutritional behavior of these athletes' diets. Only one study on the diet quality in Brazilian athletes with disabilities was found after a careful database search. The study evaluated 20 athletes during a training camp using the BHEI-R.

  1. Lines 254-255: You reported: “Unfortunately, the protocol we developed could not be adapted to apply the ADI to our data”. What do you actually mean? Could you please explain better?

A: We have completed the phrase to make it clear. 

Lines 325-327. Unfortunately, the protocol we developed could not be adapted to apply the ADI to our data because the ADI requires information on training load, which was not collected in this study.

  1. Line 275 (a sample related to point 2 above): It could be better to replace the word “But” with the word “However,”.

A: The text was adjusted.

Reviewer 2 Report

Dear Authors,

Congratulation for your effort to improve the knowledge about specific sport nutrition in Paralympic athletes, it is an area with to few studies.

We are on the opinion that your article deserves to be published, but momently it is not in condition because your methodology is not sufficiently clarified. In the results, discussion and conclusion parts we understand that your methodology is probably good, but you must do a pedagogical effort to help everybody understand what you did, with some efforts to improve the language.

Most important aspect: If Rec1 is easy to understand, it is quite impossible do understand what is the difference between original and usual diet. Please reformulate the methodology clarifying those aspects.

Others concerns to be clarified:    

line 66 “Exclusion criteria were athletes with intellectual and severe visual disabilities for which non-standardized 24-hour recall exists for this group, and novice athletes with disabilities.” Please rewrite because it is not clear what is the idea.

Line 85 “For the collection of the 24-hour recall, the “five-step multiple pass” method was used [11]” Please make in few lines a pedagogical explanation of in what it consists.

Line 88 “Quantification of the nutrient content of food recall data was performed with the Nutrition Data System for Research software (NDSR)” where did you use those data’s in the present study? Using the RDA DRIS recommendation’s does make sense to evaluate the quality of the diet in this population and compare to the results of the indexes that you used? Maybe it is interesting to provide those results in the study. And if you decide not using please justify why and eventually refer that in the limits of the study

Line 95 “To calculate the usual intake the Multiple Source Method (MSM) (Department of Epidemiology of the German Institute of Human Nutrition Potsdam-Rehbrücke, 2008–2011) was used [15].” Please make in few lines a pedagogical explanation of in what it consists, and also to help understand the concept of usual intake in relation to original.

Line 101 “The BHEI-R consists of twelve components (range between 0 to 101 100), nine of which belong to the food groups of the Brazilian Food Guide 2006, two nutrients (sodium and saturated fat), and SoFAAS (energy from solid fat, alcohol, and added 103 sugar).” Please reformulate.

Line 114 “2.3.2. Global Diet Quality Score (GDQS)” It is necessary to explain how the GDQS e was adapted to Brazilian language! And if it was only on the basis of your translation, highlighting the limits of this procedure.

Line 115 (supplementary materials S2) what is this? Where it is?

If you clarify al those point, after the results and discussion will be easier to interpret and understand. We wish you success in the reformulation of the article.

Regards

Author Response

  1. Most important aspect: If Rec1 is easy to understand, it is quite impossible do understand what is the difference between original and usual diet. Please reformulate the methodology clarifying those aspects.

A: The original value is explained on page 4, and it is the observed mean from four recalls, with no intra-individual variance correction. The usual value consists of the adjusted measurement error distribution. The original and usual differ in their percentile and SD values as shown in Figure 1. The extreme values are increased in the lower tail (5 percentile) and decreased in the higher tail (95 percentile) of the group distribution. This aspect was included in the manuscript to make it clear. Lines 192-199. During the shrinkage process, extreme values in the score/index distribution are adjusted. Specifically, the lower tail (5th percentile) is increased, while the higher tail (95th percentile) is decreased [21,22]. This process is further supported by the reduction in standard deviation (SD) observed in the usual values compared to the original and Rec1 values. To account for measurement errors, we applied corrections to each food group or dietary item in both analyzed indexes. Subsequently, the usual values were used to calculate the final usual score for BHEI-R and GDQS (Figure 1).

Others concerns to be clarified:

  1. line 66 “Exclusion criteria were athletes with intellectual and severe visual disabilities for which non-standardized 24-hour recall exists for this group, and novice athletes with disabilities.” Please rewrite because it is not clear what is the idea.

A: The text was adjusted. Lines 76-78. Athletes with intellectual and severe visual disabilities, for whom there is no standardized 24-hour recall available, as well as novice athletes with disabilities, were excluded from the study.

  1. Line 85 “For the collection of the 24-hour recall, the “five-step multiple pass” method was used [11]” Please make in few lines a pedagogical explanation of in what it consists.

A: Line 95-114. To collect the 24-hour recall data, the five-step multiple pass method (MPM) was employed [12]. Additionally, a support kit consisting of utensils (cups, glasses, and cutlery) and a food portion photography guide [13] were utilized. The MPM consists of five stages. The first stage involves the uninterrupted listing of the foods and beverages consumed in the past 24 hours. The second stage is based on a list of commonly forgotten foods for each meal. In the third stage, the interviewee provides the consumption time for each food/supplement or beverage, detailing the location and occasion of consumption. The fourth stage, known as the elaboration stage, requires the interviewee to provide additional details about the meals, including quantities of the consumed foods, brands, and preparation methods. At this point, household measures such as teacups, cups, tablespoons, teaspoons, etc., are recorded. In the fourth stage, information regarding the meals time and occasion are reviewed. The fifth and final stage is the final review, where a thorough examination of the already reported foods is conducted, and any possible forgotten or unreported foods are included [12].

The information from the 24-hour recall was registered in food portion size and units and converted into grams of consumed foods using the standardized quantities from the photography guide [13] and a Brazilian household measurement table [14].

  1. Line 88 “Quantification of the nutrient content of food recall data was performed with the Nutrition Data System for Research software (NDSR)” where did you use those data’s in the present study?

A: We rewrote the sentence to be more specific. Lines 114-118. To quantify the content of food groups and dietary items from the food recall data, the Nutrition Data System for Research software (NDSR) developed by the Nutrition Coordinating Center at the University of Minnesota, USA, was utilized [15]. Brazilian food items and recipes were either directly entered in the NDSR or adapted from foods with similar nutritional profiles [16].

Using the RDA DRIS recommendation’s does make sense to evaluate the quality of the diet in this population and compare to the results of the indexes that you used? Maybe it is interesting to provide those results in the study. And if you decide not using please justify why and eventually refer that in the limits of the study.

A: The use of RDA is limited to evaluation of individual dietary planning and should not be used to evaluate dietary intake of groups or population [IOM 2000, Murphy et al 2006; doi:   10.1016/j.jada.2006.08.021] as is the case of our study. We evaluated food group intake and not nutrients. The evaluation of nutrients related to the EAR, AI and UL from this population was previously evaluated in a previous publication and cited in the manuscript [20] (Sasaki et al 2021, doi:10.1016/j.nut.2020.110992.)

  1. Line 95 “To calculate the usual intake the Multiple Source Method (MSM) (Department of Epidemiology of the German Institute of Human Nutrition Potsdam-Rehbrücke, 2008–2011) was used [15].” Please make in few lines a pedagogical explanation of in what it consists, and also to help understand the concept of usual intake in relation to original.

A: Lines 125-136.  In short, the MSM applies a three-step methodology to correct intra-individual intake errors. The first step considers the intake probability using a logistic regression calculation. The second step takes the values of the observed intake group data from each 24h recall. A linear regression model was applied, and the values were modeled, and the corresponding residuals of the linear regression transformed to normality/symmetry by a two-parameter Box-Cox transformation. Then the inter- and intra-individual variance were determined, and the intra-individual variance removed, which resulted in a shrinkage of the residuals distribution. The quantity calculated in this shrinkage process for each person was then transformed back to the original scale. The third step, which is the distribution of usual intake value for the study population, was estimated by multiplication of the results of steps 1 and 2 [17,18].

  1. Line 101 “The BHEI-R consists of twelve components (range between 0 to 101 100), nine of which belong to the food groups of the Brazilian Food Guide 2006, two nutrients (sodium and saturated fat), and SoFAAS (energy from solid fat, alcohol, and added 103 sugar).” Please reformulate.

A: The text was adjusted. Line 143-148. The BHEI-R ranges from 0 to 100, and is composed of 12 components, which include nine food groups from the Brazilian Food Guide 2006, two nutrients (sodium and saturated fat), and SoFAAS (energy from solid fat, alcohol, and added sugar). The portions of the nine food groups and sodium are presented as energy density (g / 1000 total energy intake), whereas saturated fat, and SoFAAS are presented in percentage of total energy intake (TEI).

  1. Line 114 “2.3.2. Global Diet Quality Score (GDQS)” It is necessary to explain how the GDQS e was adapted to Brazilian language! And if it was only on the basis of your translation, highlighting the limits of this procedure.

A: There were no need to adapt the GDQS to Portuguese as it is based on the food reported in the 24-hour recalls. All the 24-hour recalls were done in Portuguese and the GDQS portion sizes are based on a daily intake.

  1. Line 115 (supplementary materials S2) what is this? Where it is?

A: This is the supplementary table. All were attached together with the other documents during the submission of the article.

If you clarify al those point, after the results and discussion will be easier to interpret and understand. We wish you success in the reformulation of the article.

Round 2

Reviewer 2 Report

Dear Authors,

congratulation, you succeed the reformulation

We wish you success!

Author Response

Thank you for your comments.